# Long-Read Sequencing Reveals Cell- and State-Specific Alternative Splicing in 293T and A549 Cell Transcriptomes

**DOI:** 10.3390/ijms27010487

**Published:** 2026-01-03

**Authors:** Xin Li, Hanyun Que, Zhaoyu Liu, Guoqing Xu, Yipeng Wang, Zhaotong Cong, Liang Leng, Sha Wu, Chunyan Chen

**Affiliations:** 1Institute for Chinese Medicine Frontier Interdisciplinary Science and Technology, Shaanxi University of Chinese Medicine, Xianyang 712046, China; xinli@cdutcm.edu.cn; 2Institute of Herbgenomics, Chengdu University of Traditional Chinese Medicine, Chengdu 611137, China; qhy@stu.cdutcm.edu.cn (H.Q.); liuzhaoyu2008@hotmail.com (Z.L.); xuguoqing_0201@163.com (G.X.); congzt@cdutcm.edu.cn (Z.C.); lling@cdutcm.edu.cn (L.L.); wusha19960801@163.com (S.W.); 3School of Basic Medical Sciences, Chengdu University of Traditional Chinese Medicine, Chengdu 611137, China; 4Department of Biopharmaceutical Sciences, College of Pharmaceutical Sciences, Soochow University, Suzhou 215123, China; yipengwang@alu.suda.edu.cn; 5School of Pharmacy, Chengdu University of Traditional Chinese Medicine, Chengdu 611137, China

**Keywords:** 293T, A549, RNA-seq, differentially expressed genes (DEGs), genes with differential transcript usage (gDTUs), FLAIR, SQANTI3

## Abstract

Alternative splicing (AS) is a fundamental mechanism governing transcriptomic diversity and cellular identity. Although 293T (human embryonic kidney) and A549 (human lung adenocarcinoma) cell lines are widely used, cell-type-specific splicing dynamics—including responses to receptor overexpression—remain incompletely characterized. To address this, we integrated Oxford Nanopore long-read sequencing with BGI short-read data to profile transcriptomes under both basal and GPCR-overexpressing conditions (*ADORA*3 in 293T; *P2RY12* in A549). Full-length isoform analysis using FLAIR and SQANTI3 revealed extensive transcriptomic complexity, including 18.02% novel isoforms in 293T and 19.52% in A549 cells. We found that 293T cells exhibited a stable transcriptome architecture enriched in splicing-related pathways, whereas A549 cells underwent broader transcriptional remodeling linked to tumorigenic processes. These findings suggest that 293T cells may be a suitable model for investigating splicing regulation, while A549 cells could serve as a relevant system for exploring tumor-related transcriptome dynamics. Our work elucidates context-dependent AS regulation and underscores the value of integrating long-read sequencing with FLAIR/SQANTI3 for dissecting cell-state-specific transcriptome dynamics.

## 1. Introduction

Alternative splicing (AS) is a key mechanism for expanding transcript and protein diversity in eukaryotic systems. AS contributes to fine-tuned regulation of cellular identity, signaling, and adaptability by enabling a single gene to produce multiple mRNA isoforms with distinct, and sometimes opposing, functions. Its dynamic control arises from the interaction between cis-regulatory RNA elements and trans-acting splicing factors, shaped by cell type, developmental stage, and environmental context. Disruptions in splicing regulation have been implicated in a wide range of human diseases, including cancer, neurodegeneration, and immune disorders [1]. Despite its importance, many splicing events remain under-characterized, particularly in perturbed or non-physiological conditions.

Emerging evidence implicates cell signaling pathways mediated by G protein-coupled receptors (GPCRs) in the regulation of alternative splicing. GPCR activation is known to induce broad transcriptional and post-transcriptional reprogramming. Downstream components, such as β-arrestin, have been shown to modulate splicing factor activity and spliceosome function, particularly through phosphorylation of serine/arginine-rich (SR) proteins in specific cellular contexts [2]. While these observations point to a plausible link between extracellular signaling and splicing control, systematic studies on GPCR-driven alternative splicing remain limited. A key unresolved question is whether these splicing effects are cell-type-specific, especially within the common overexpression models routinely used in signaling research [3].

To address this gap, we focused on two specific GPCRs: *ADORA3* in HEK293T cells and *P2RY12* in A549 cells. This selection was based directly on our group’s prior work characterizing the transcriptomic impact of plasmid transfection and GPCR overexpression in these lines. We previously documented alterations in gene expression and splicing following lentiviral *ADORA3* overexpression in HEK293T cells and plasmid-mediated overexpression of GFP-tagged *P2RY12* in A549 cells [4,5]. Building on these established models, this study leveraged a combined long-read and short-read sequencing approach to perform a comparative transcriptomic analysis under both native and GPCR-overexpressing conditions, aiming to systematically identify cell-type-specific and shared splicing responses.

The 293T (human embryonic kidney) and A549 (human lung adenocarcinoma) cell lines are widely utilized in biomedical research due to their robust growth and high transfection efficiency. 293T cells are a standard model for recombinant protein production and viral packaging, while A549 cells serve as a cornerstone in cancer biology for studying drug responses and metastatic mechanisms [6,7]. Despite their prevalent use, the baseline alternative splicing landscapes of these lines—and how they respond to common experimental perturbations, such as gene overexpression—remain inadequately characterized. This gap complicates the interpretation of transcriptomic data from experimental manipulations in these systems.

A key technical limitation has been the reliance on short-read RNA sequencing, which often fails to resolve full-length isoform structures and can miss complex splicing events, such as recursive or distant exon junctions [8]. To overcome this, we employed long-read sequencing via Oxford Nanopore Technologies (ONT), which allows for the direct sequencing of intact RNA molecules, enabling the comprehensive detection of novel isoforms and complex splicing patterns [9].

In this study, our previously published datasets were integrated with new analyses to compare the transcriptomes of 293T and A549 cells under both native and GPCR-overexpressing conditions. Dedicated bioinformatic pipelines were applied to characterize isoform diversity and to identify splicing events and unannotated transcripts associated with receptor overexpression. This work aimed to characterize cell-type-specific splicing patterns in two commonly used experimental models, employing long-read sequencing to investigate the transcriptomic alterations that may accompany genetic perturbations.

## 2. Results

### 2.1. Full-Length Transcriptome Profiles of 293T and A549 Cells

We employed an integrated RNA sequencing strategy combining BGI short-read and ONT long-read platforms to comprehensively characterize the full-length transcriptomes of 293T and A549 cells. The study samples consisted of unstimulated 293T and A549 cells, along with 293T cells that overexpress *ADORA3* and A549 cells that overexpress *P2RY12*. Raw ONT reads were first processed using FLAIR to collapse and merge transcripts (Figure 1). We classified all high-confidence transcript isoforms identified using the SQANTI3 tool based on the GENCODE v48 reference annotation. The specific categories are as follows: full-splice match (FSM), which refers to isoforms that exactly match all splice junctions of an annotated reference transcript; incomplete-splice match (ISM), where isoforms align with a reference transcript’s splice junctions but show truncations or extensions at either the 5’ or 3’ end; novel in catalog (NIC), denoting isoforms that are new combinations of known, annotated splice junctions; and novel not in catalog (NNC), consisting of isoforms containing at least one new splice junction not present in the reference annotation. Additionally, antisense, fusion, genic, genic intron, and intergenic transcripts were grouped into the ‘other’ category (Figure 2A). Representative genes expressing FSM, ISM, NIC, and NNC isoforms were visualized using Integrative Genomics Viewer (IGV) to illustrate SQANTI3 classification structures (Figure 2B). Isoforms identified within both the NIC and NNC categories are novel, as they are not recorded in the GENCODE v48 annotation. As shown in Figure 2C, both NIC and NNC are present in the two cell lines, although the difference in their transcriptome annotation consistency is small. A comprehensive summary of QC metrics for all identified novel isoforms (both NIC and NNC) is now provided (Appendix A).

Each unique combination of exons defines a specific mRNA isoform, diversifying the transcriptional output of a gene. These variant mRNAs, in turn, serve as templates for the synthesis of distinct protein isoforms, ultimately contributing to a rich array of functional proteoforms [10]. We assessed isoform counts per gene to further examine transcript diversity across genes. In 293T cells, 6710 genes expressed a single isoform, 4504 genes expressed 2–3 isoforms, 1733 genes expressed 4–5 isoforms, and 1387 genes expressed ≥6 isoforms. A549 cells showed similar trends: 6709 genes with 1 isoform, 4155 genes with 2–3 isoforms, 1432 genes with 4–5 isoforms, and 1178 genes with ≥6 isoforms (Figure 2D). Furthermore, 293T cells harbored more genes with ≥6 isoforms (1387 vs. 1178). This observed difference, though noted within broadly similar overall distributions, is consistent with a higher transcriptional complexity in 293T cells. In both lines, isoform counts per gene approximately followed a long-tail distribution, with much of the transcript diversity concentrated in a limited subset of genes that produce numerous isoforms. The variation in high-complexity genes might reflect cell-type-specific differences in splicing regulation, which could be linked to distinct developmental, functional, or pathological contexts. Together, these observations provide a basis for considering how transcript diversity may be organized and regulated across different cell types.

Given the discrepancies in transcript isoform classification criteria between the two major human gene annotation databases, GENCODE and RefSeq (e.g., isoforms defined as NIC or NNC in GENCODE may be classified as FSM in RefSeq), we performed a cross-database comparative analysis. In addition, all non-FSM isoforms under the GENCODE v48 annotation were compared against the RefSeq-curated annotations. The results showed that in 293T cells, out of 10,535 non-FSM isoforms included in the analysis, 1594 were redefined as FSM in RefSeq (Figure 2E). Furthermore, the two databases also differed in their assignment of novel isoforms: GENCODE annotated 5194 NIC and 3210 NNC isoforms, whereas RefSeq annotated 2474 NIC and 4511 NNC isoforms. Correspondingly, in A549 cells (Figure 2F), 1321 out of 10,024 non-FSM isoforms were classified as FSM in RefSeq; GENCODE annotated 4438 NIC and 3364 NNC isoforms, while the corresponding numbers in RefSeq were 2147 and 4435, respectively. The discovery of thousands of novel isoforms prompts us to consider the reason for their extensive presence in the human genome. The gene age problem might be a contributing cause. Evolutionarily ancient and conserved genes tend to be more extensively studied; consequently, their isoforms are generally better characterized than those of more recently evolved genes [11,12,13].

To investigate how gene evolutionary age influences annotation inconsistency, we analyzed the gene age distribution of protein-coding genes that harbor isoforms inconsistently annotated between GENCODE and RefSeq in 293T and A549 cells. As shown in Figure 2G, the vast majority of these genes were evolutionarily ancient, non-primate-specific genes (branch 0–7). In this context, 4260 out of 17,887 total genes in branch 0–7 in 293T cells contained such isoforms, compared with only 43 out of 818 primate-specific genes (branch 8–14). In this gene age classification system, branches 0–7 represent genes that originated before the primate lineage, whereas branches 8–14 correspond to primate-specific genes.

### 2.2. Transcriptome Expression Under Overexpression Intervention

Based on the comprehensive transcriptomic characterization, we further performed principal component analysis (PCA) to evaluate the global transcriptional responses induced by receptor overexpression. PCA revealed distinct clustering patterns, clearly separating 293T cells from A549 cells. Similarly, a clear separation was observed between *ADORA3*-overexpressing 293T cells and *P2RY12*-overexpressing A549 cells, suggesting that the distinct cellular contexts drove major transcriptomic differences (Figure 3A).

The Venn diagram shows the overlap between two sets of DEGs. One set represents the DEGs between 293T and A549 cells under baseline conditions, while the other set represents the DEGs between 293T cells overexpressing *ADORA3* and A549 cells overexpressing *P2RY12*. Analysis revealed a significant overlap between the two DEG sets, comprising a total of 6638 shared genes. This finding indicates that, despite different genetic perturbations, the cells share numerous consistently differentially expressed genes, suggesting that these commonly altered genes may be involved in core biological processes and hold important research significance (Figure 3B). We performed a comparative analysis of DEGs for the following comparisons: 293T versus A549 cells, and *ADORA3*-overexpressing 293T versus *P2RY12*-overexpressing A549 cells. The 293T versus A549 comparison yielded 5662 upregulated and 5275 downregulated genes, respectively. Similarly, the comparison of the two receptor-overexpressing groups identified 5363 upregulated and 5125 downregulated genes, respectively (Figure 3C).

### 2.3. Functional Changes Under Overexpression Intervention

To gain functional insights, we performed an integrated Gene Ontology enrichment and Kyoto Encyclopedia of Genes and Genomes (KEGG) pathway analysis on the DEG profiles from 293T and A549 cells, including profiles from their respective overexpression conditions. Figure 3D presents the GO enrichment analysis for molecular functions in the 293T and A549 cell lines. Significantly enriched terms included DNA-binding transcription activator activity (RNA polymerase II-specific), actin binding, microtubule binding, organic acid binding, carboxylic acid binding, extracellular matrix structural constituents, integrin binding, and collagen binding. The terms for DNA-binding transcription activator activity (RNA polymerase II-specific) and actin binding were highly significant (*p* = 1.0 × 10^−5^), as was microtubule binding (*p* = 2.0 × 10^−5^). Our analysis revealed differences in the cellular characteristics of the two lines. These variations encompassed multiple biological domains, including, but not limited to, transcriptional regulation, cytoskeletal organization, and extracellular matrix interactions. The differences observed in the A549 lung cancer cell line may be implicated in tumor-specific processes. KEGG pathway enrichment analysis revealed distinct functional profiles between the 293T and A549 cell lines (Figure 3E). Several key pathways were robustly enriched, including the PI3K-Akt signaling pathway (*p* = 1.0 × 10^−5^), which is critical for cell survival, proliferation, and metabolism. Additionally, pathways associated with cytoskeletal organization and adhesion were prominent, such as focal adhesion (*p* = 1.0 × 10^−5^), ECM–receptor interaction (*p* = 0.02), and cell adhesion molecules (*p* = 2.0 × 10^−5^), all of which contribute to cell migration and extracellular matrix interactions. These may be relevant to the invasive potential of A549 lung cancer cells. Significant enrichment was also observed in the complement and coagulation cascades (*p* = 3.0 × 10^−5^), suggesting differences in immune and inflammatory responses. Other relevant pathways, such as retinol metabolism and axon guidance, pointed to metabolic and neurodevelopmental disparities. Collectively, these results highlight pronounced differences in signal transduction, cell adhesion, extracellular matrix remodeling, and immune-related pathways between the two cell lines, reflecting their distinct biological contexts. The enhanced activity of adhesion, migration, and immune evasion pathways in A549 cells provides valuable insights into its tumor-specific behavior.

Following overexpression of *ADORA3* and *P2RY12* in 293T and A549 cell lines, respectively, GO enrichment analysis of molecular functions revealed systematic differences between the two cell models (Figure 3F). Significant enrichment of “DNA-binding transcription activator activity” indicated distinct mechanisms in gene expression regulation. Additionally, terms such as “actin binding”, “extracellular matrix structural constituent”, and “integrin binding” suggested differences in cytoskeletal organization and extracellular matrix interactions, potentially influencing cell adhesion and motility. Altered signaling capacity was reflected in the enrichment of “transmembrane receptor protein tyrosine kinase activity” and “insulin-like growth factor binding,” which may affect proliferative and metabolic responses, while “enzyme inhibitor activity” pointed to possible divergence in metabolic regulation. Collectively, these functional distinctions highlight fundamental differences in the biological characteristics of *ADORA3*- and *P2RY12*-overexpressing cell lines, providing critical insights for further investigation into their cell-type-specific behaviors.

As shown in Figure 3G, KEGG pathway enrichment analysis of the *ADORA3* and *P2RY12* cell lines revealed the following key functional differences. In terms of signal transduction, significant enrichment of the PI3K-Akt and MAPK signaling pathways suggested differences between the two cell lines in proliferation, survival, and response to external signals, which may be related to their growth characteristics or potential oncogenic mechanisms. Among cytoskeleton and adhesion-related pathways, significant enrichment of “cytoskeleton in muscle cells”, “focal adhesion” and “ECM–receptor interactions” reflected differences in cell motility, morphology maintenance, and adhesion functions, which may influence their invasive capacity and tissue repair potential. In immune and inflammatory regulation, enrichment of the complement and coagulation cascades indicates potential differences in immune response and inflammatory modulation mechanisms. Additionally, the axon guidance pathway may suggest a possible association with nervous system function, while the cardiomyopathy and retinol metabolism pathways, although limited in information, may reflect differences in metabolic characteristics or tissue-specific functions between the two cell lines.

In summary, systematic differences exist between *ADORA3-*overexpressing 293T and *P2RY12-*overexpressing A549 cells across multiple key biological pathways, including signal transduction, cell adhesion and motility, immune regulation, and metabolism. These findings provide pathway-level insights for elucidating their functional characteristics and potential physiological and pathological significance.

### 2.4. Alternative Splicing Events Under Overexpression Intervention

AS serves as a critical mechanism for expanding proteomic diversity and precisely regulating gene expression [14]. In this study, we systematically investigated AS events in *ADORA3*-overexpressing 293T cells and *P2RY12*-overexpressing A549 cells to delineate the remodeling effects of overexpression intervention on the splicing landscape. We performed an analysis of gDTUs using SUPPA in 293T and A549 cells, both under baseline conditions and following overexpression. AS events were categorized into the following types: skipped exon (SE), alternative 5′/3′ splice sites (A5/A3; identified using the splice site option “SS”), mutually exclusive exons (MX), retained intron (RI), and alternative first/last exons (AF/AL).

Genes with differential transcript usage (gDTU) analysis comparing transcriptomes before and after overexpression stimulation identified genes exhibiting significant differential transcript usage. These changes were often driven by specific alternative splicing (AS) events, such as exon skipping or intron retention. Under baseline conditions, gDTUs analysis in 293T and A549 cells identified widespread alternative splicing events across multiple categories. Under baseline conditions, the most frequent event type was AF (10,385 AS events involving 3664 genes), followed by SE (5250 events; 3724 genes), A5 (2490 events; 1901 genes), A3 (2428 events; 1953 genes), AL (4387 events; 1896 genes), RI (1390 events; 1019 genes), and MX (747 events; 578 genes). In addition, the number of AS events generally increased across most categories in the overexpression models (293T-*ADORA3* and A549-*P2RY12*). AF remained the most abundant (11,128 events; 3885 genes), followed by SE (5629 events; 3935 genes), AL (4661 events; 1955 genes), A5 and A3 (both 2563 events; affecting 1956 and 2054 genes, respectively), MX (845 events; 667 genes), and RI (1289 events; 976 genes) (Figure 4A,B).

GO enrichment analysis identified distinct molecular function profiles in gDTUs between 293T and A549 cells (Figure 4C) and between *ADORA3* and *P2RY12*-related cells (Figure 4D). In the 293T vs. A549 comparison, significant enrichments were observed for DNA-binding transcription factor activity, GTPase regulator activity, histone-modifying activity, and protein serine/threonine kinase activity (adjusted *p*-value < 0.05). These patterns likely reflect differences in transcriptional regulation, signal transduction, and epigenetic modification between the cell lines. Additionally, enrichment of genes related to ubiquitin-like protein ligase binding may indicate variations in protein stability regulation.

In the *ADORA3* vs. *P2RY12* comparison, notable enrichments included protein serine/threonine kinase activity, protein serine kinase activity, and catalytic activity acting on DNA. These findings suggest potential differences in gene expression regulation, signal transduction, protein homeostasis, and epigenetic modification between the two cell types. Overall, these results provide insights into the molecular distinctions between the cell lines and highlight areas for further exploration of their biological mechanisms.

Figure 4E,F display the KEGG pathway enrichment analysis results for 293T versus A549 cells and *ADORA3* versus *P2RY12*-related cells, respectively. These analyses identified both differences and commonalities in several key biological pathways across the cell lines. For instance, the enriched pathways in both comparisons included neurodegenerative diseases (e.g., Alzheimer’s and Parkinson’s diseases), COVID-19, thermogenesis, nucleotide metabolism, and, in the case of 293T vs. A549, ubiquitin-mediated proteolysis. These findings may reflect potential variations in how the cell lines handle neurodegenerative mechanisms, energy metabolism, DNA repair, and responses to viral infections. Overall, these results provide useful insights that could inform further studies into the functional distinctions and underlying biological mechanisms between these cell types.

Based on SUPPA analysis, Figure 4G profiles genes concurrently identified as gDEGs and gDTUs in the 293T and A549 cell lines, providing a breakdown by local splicing event type (SE, A5, A3, MX, RI, AF, and AL). SE events were most frequent (1553 genes), followed by RI (256 genes), while MX events were the least common (64 genes), indicating that exon skipping and intron retention represent key post-transcriptional mechanisms potentially contributing to proteomic diversity. The distinct splicing profiles reflect transcriptional regulatory heterogeneity between the cell lines, with the prevalence of SE and RI events underscoring their central role in maintaining gene expression plasticity, possibly supporting cellular adaptation to specific physiological or pathological contexts. These findings systematically characterize splicing divergence between A549 and 293T cells, providing a foundation for investigating cell-type-specific gene regulation, disease mechanisms, and therapeutic targets, with SE and RI events warranting particular focus in future studies. Figure 4H displays the distribution of genes identified as both gDTUs and gDEGs across different local splicing event types in 293T cells overexpressing *ADORA3* and A549 cells overexpressing *P2RY12*. The specific gene counts are as follows: 276 genes in A3 events, 222 in A5 events, 786 in AF events, 368 in AL events, 76 in MX events, 230 in RI events, and 1586 in SE events. SE events represent the most predominant category, indicating that exon skipping serves as the primary splicing alteration mechanism, likely playing a critical role in regulating gene expression and generating protein diversity. The substantial proportions of AF and AL events suggest notable variability in transcriptional initiation and termination regions, which may influence gene function and expression levels. In contrast, MX events were the least frequent, reflecting their higher specificity.

These findings delineate the splicing profiles of the two cell lines under specific gene overexpression conditions, providing valuable insights into gene regulatory mechanisms, cell-specific functions, and disease-related pathways, while also laying the groundwork for future investigations into the role of key splicing events in gene function regulation. These results systematically outline how *ADORA3* and *P2RY12* overexpression influences transcript usage across fundamental biological processes and disease-relevant mechanisms, with particularly strong implications for neurodegeneration and cancer biology.

### 2.5. IGV Visualization of Alignment Data Uncovered Differences in Gene Expression and Splicing Regulation

We performed a comparative analysis of alignment data between 293T and A549 cell lines, incorporating read coverage, splice junctions, and gene annotation information (Figure 5 and Appendix A). As shown in Figure 5A, IGV visualization of a specific locus on chromosome 12 (chr12: 52,282,552–52,293,544) illustrated the alignment profiles of both cell lines. Read coverage is displayed with darker shades, representing higher sequencing depth, and splice junctions are depicted as arching lines, indicating RNA-seq splicing events. Gene annotation tracks (ENSG00000205426.gtf) are provided at the bottom, showing gene and transcript structures, with exons as blue blocks, introns as gray connecting lines, and specific transcript IDs labeled.

Interestingly, at this locus, while the reference gene KRT81 (ENST00000327741.9) is annotated in both GENCODE and RefSeq as containing nine exons, our long-read sequencing data identified a previously unannotated shorter isoform consisting of only five exons. This novel isoform was consistently supported by RNA-seq data in both 293T and A549 cell lines. Comparative analysis further revealed distinct gene expression and splicing patterns between the two cell lines, providing important insights into cell-type-specific transcriptional regulation.

In addition, we compared the alignment data between native A549 and *P2RY12*-overexpressing A549 *(P2RY12*-1) cells. Figure 5B displays the IGV visualization of the same genomic region, using the same representation for read coverage, splice junctions, and gene annotations. Comparative examination revealed discernible alterations in read coverage and splicing patterns following *P2RY12* overexpression, suggesting receptor-induced changes in gene expression or splicing regulation. These findings offer valuable mechanistic clues for further functional studies.

## 3. Discussion

AS is a major mechanism contributing to transcriptomic complexity and functional diversity [15]. However, its cell-type-specific characteristics, especially under genetic perturbations such as gene overexpression, remain insufficiently explored in commonly used cell models. In this study, we employed long-read RNA sequencing combined with FLAIR and SQANTI3 pipelines to profile the full-length transcriptomes of 293T and A549 cells under basal and GPCR-overexpressing conditions. Our findings reveal that while both *ADORA3*- and *P2RY12*-overexpressing cells exhibited significant global transcriptomic reprogramming, the nature and extent of these responses differed between cell types. Our analysis revealed a substantial occurrence of alternative first/last exons (AF/ALs), which contradicts previous findings suggesting skipped exons (SEs) as the most prevalent alternative splicing (AS) event in human genomes [16]. Despite this, fewer genes were affected by AF/ALs compared to those impacted by SEs. Two hypotheses may account for this observation: first, the FLAIR-SQANTI3 pipeline might exhibit enhanced sensitivity to initial or terminal exon variations; second, it is plausible that AF/AL events are genuinely more frequent than SE events within cell line environments. Further research is necessary to empirically validate these postulations.

Alternative splicing has been significant in biological research since introns were discovered, primarily due to its pivotal role in fundamental processes such as gene expression regulation [17,18]. Nevertheless, research in this area is hampered by the similarity between isoforms from the same gene being confused with other factors, such as similarities among genes in the same family or other repetitive regions within genes [19]. Our prior research has demonstrated the viability of combining second- and third-generation sequencing methodologies for identifying novel plant isoforms, facilitated by the advent of third-generation sequencing in isoform characterization [20,21,22]. Similarly, third-generation sequencing has enabled the discovery of novel isoforms across diverse human tissues and cell types [23,24,25]. The identification of novel isoforms and gDTUs continues to face certain limitations. The utilization of only three replicates in this study, a consequence of resource restrictions, may influence the reliability of alternative splicing analysis, primarily because this analysis is acutely vulnerable to both biological noise and experimental batch effects. Furthermore, the inherent variability observed in ONT reads represents an often-overlooked aspect in the experimental design, a challenge that could be addressed using techniques such as spike-in RNA quantification [26,27]. Thus, while these newly identified isoforms are supported by RNA-seq data, they necessitate additional experimental validation, such as PCR verification, before proceeding with functional investigations.

293T cells showed higher concordance with GENCODE annotations, a larger proportion of FSM, and fewer novel transcripts, suggesting a more stable and well-annotated transcriptomic baseline. In contrast, A549 cells displayed greater transcriptomic novelty and a broader transcriptional response to *P2RY12* overexpression, consistent with their cancer-derived origin and inherent transcriptional plasticity [28]. DEG and gDTUs analyses revealed distinct functional signatures. In 293T cells, DEGs and gDTUs were enriched in RNA processing and spliceosome-associated functions, implying that splicing regulation is a major target of transcriptional modulation in this context.

Conversely, A549 DEGs and gDTUs were more prominently involved in pathways related to cell cycle, translation, and stress response, reflecting the proliferative and stress-adaptive nature of cancer cells [29]. Additionally, ES emerged as the dominant AS event in both cell types, but the extent of gDTUs was more pronounced in 293T cells. This suggests that 293T cells, despite their higher splicing fidelity, retain a capacity for fine-tuned splicing regulation in response to specific stimuli. The enrichment of spliceosome-related functions further positions 293T cells as a suitable model for mechanistic studies of splicing regulation. Given that this is a descriptive and exploratory study, mechanistic validation—such as RNA-binding protein (RBP) mapping or functional assays—will be required to determine the biological relevance of the identified transcript isoforms and splicing events [30].

## 4. Materials and Methods

### 4.1. Data Sources

The human lung adenocarcinoma cell line A549 was purchased from Kinlogix (Kinlogix, Guangzhou, China) and grown in RPMI1640 medium (Gibco, Waltham, MA, USA) containing 10% fetal bovine serum (Gibco, Waltham, MA, USA) and 100 µg/mL streptomycin and 100 U/mL penicillin (Beyotime, Shanghai, China). The Human Embryonic Kidney 293T cells were purchased from Procell (Procell, Wuhan, China) and grown in DMEM (Gibco, Waltham, MA, USA). The cells were incubated at 37 ◦C with 5% CO2 in an incubator.

All sequencing data used in this study are publicly available at the National Genomics Data Center (NGDC). The short-read RNA-seq datasets previously published [8,9] under BioProjects PRJCA036438 and PRJCA036439 can be accessed at the GSA-Human experiment-level pages HRA010505 (https://ngdc.cncb.ac.cn/gsa-human/browse/HRA010505, access on 21 February 2025) and HRA010506 (https://ngdc.cncb.ac.cn/gsa-human/browse/HRA010506, access on 21 February 2025), respectively, where the sequencing reads and detailed experimental information are provided. The Oxford Nanopore long-read sequencing data generated in this study have been submitted under BioProject PRJCA052561, and the corresponding submission information is available at the GSA-Human page subHRA023356 (https://ngdc.cncb.ac.cn/gsa-human/submit/hra/subHRA023356, access on 21 February 2025). All access links and experiment IDs have been updated to ensure full transparency and reproducibility.

### 4.2. Sequencing and Quality Control

For short-read RNA-seq, Illumina libraries were generated for four groups: 293T, *ADORA3* overexpression, A549, and *P2RY12* overexpression (*n* = 3 biological replicates per group). Total RNA was extracted using TRIzol, followed by rRNA depletion and library construction using the MGIEasy Circularization Kit (MGI, Shenzhen, China). Sequencing was performed on an Illumina platform (DNBSEQ-T7) to obtain 150 bp paired-end reads. Quality assessment was conducted using FastQC (v0.12.1) and Trimmomatic (v.0.39), with all samples showing a high base accuracy (Q30 > 93%) and an adequate sequencing depth for differential expression analysis [31]. These short-read data complemented the long-read quantification and supported the validation of differential expression findings. For long-read transcriptome profiling, sequencing was performed on the PromethION platform following the manufacturer’s protocol. Base-calling was carried out with Guppy, and low-quality Nanopore reads (mean Phred quality ≤ 7) were removed before downstream analysis. Across the 12 libraries, we obtained approximately 4.4–6.5 million long reads per sample, with N50 values ranging from 1.3 to 1.7 kb and mean Q scores of 16–17 after filtering.

### 4.3. Short-Read RNA Sequencing and Analysis

Short reads were aligned using STAR (version 2.7.10b, https://github.com/alexdobin/STAR, access on 21 February 2025) to generate splice junction files for subsequent correction of long-read sequencing data. Transcript quantification was performed with Kallisto (version 0.50.1, https://github.com/pachterlab/kallisto, access on 21 February 2025). The kallisto index was built based on the GENCODE v48 annotation (https://ftp.ebi.ac.uk/pub/databases/gencode/Gencode_human/release_48/gencode.v48.annotation.gtf.gz, access on 21 February 2025) and the human genome reference (GRCh38.p14, https://ftp.ebi.ac.uk/pub/databases/gencode/Gencode_human/release_48/GRCh38.p14.genome.fa.gz, access on 21 February 2025). Quantification was conducted using the kallisto quant module [32].

### 4.4. Isoform Reconstruction and Classification with FLAIR and SQANTI3

Full-length alternative isoform analysis of RNA (FLAIR), developed by Angela Brooks’s team at the University of California, Santa Cruz, is a computational tool designed for the identification of high-confidence transcripts, analysis of alternative splicing events, and detection of differential transcript isoform expression [33]. Long-read RNA sequencing data were processed using the FLAIR pipeline (version 2.0) following the recommended workflow. First, Illumina short reads from all samples were aligned to the hg38 reference genome using STAR with sjdbOverhang = 149, and splice junction files (SJ.out.tab) from each sample were merged to generate a comprehensive high-confidence splice junction set for downstream correction. Nanopore long reads were aligned to the hg38 genome using flair align with default minimap2 parameters. Splice site correction was performed using flair correct, which incorporates both the STAR-derived splice junctions and the NCBI RefSeq gene annotation to refine long-read splice boundaries.

Transcriptome assembly was conducted using flair collapse, with several stringent options enabled, including --stringent, --check_splice, and --annotation_reliant, to ensure high-confidence isoform structures and reduce potential false-positive novel junctions. Read-to-transcript assignment files were generated with --generate_map.

Isoform quantification across samples was performed using flair quantify based on the unified transcriptome reference. Differential transcript expression was assessed using flair diffExp, whereas differential alternative splicing was tested using flair diffSplice with the --test option. All analyses were performed on the merged transcriptome and consistent sample-specific count matrices [34,35]. SQANTI3 is a tool specifically designed for the quality control, curation, and annotation of long-read transcript models obtained from third-generation sequencing technologies. SQANTI3 calculates quality descriptors for transcript models, junctions, and transcript ends using its annotation framework [36]. Transcriptome filtering and quality control were performed using SQANTI3 (v5.5) to assess long-read-derived transcript models based on structural and sequence features. Isoforms were first classified via alignment to the GENCODE v48 reference annotation. Transcripts exhibiting full-splice site concordance with a reference transcript were categorized as FSMs. The remaining non-FSM isoforms—including ISM, NIC, NNC, genic, genic_intron, intergenic, antisense, and fusion transcripts—were subsequently re-annotated using the manually curated RefSeq GTF (release: 27 August 2024) to improve structural accuracy and aid in the discrimination of genuine novel isoforms. All non-FSM categories were consolidated into a high-confidence novel isoform set for downstream analysis.

Quality filtering was performed using the SQANTI3 QC module under default parameters, with the addition of the -c argument to incorporate coverage information. To minimize artifacts, non-canonical splice sites (non-GT/AG) were retained only if supported by cross-sample splice junction evidence and at least three uniquely mapped long reads in at least one sample. For FSMs, the SQANTI3 filter and rescue modules were applied to retain splice sites supported by limited long-read evidence, thereby improving annotation sensitivity while maintaining accuracy [37]. The final output comprised a high-confidence, cell-type-specific transcriptome, with robust structural annotation and quality metrics suitable for subsequent isoform-level expression and differential usage analysis [38].

### 4.5. Alternative Splicing Identification and Differential Analysis

Alternative splicing events were identified and quantified using SUPPA2 (v2.3) [39]. To enhance analytical rigor, we applied stringent filtering criteria to the alternative splicing events identified by SUPPA2, as detailed below: First, to ensure reliability across biological replicates, each alternative splicing event was required to be detected in at least two out of the three biological replicates. This consistency filter was implemented using an in-house script. Second, to retain events with substantial biological differences, statistical thresholds were applied, where only events meeting |ΔPSI| ≥ 0.1 and *p*-value < 0.05 were retained. This dual-filtering strategy was designed to ensure both the reliability and the statistical significance of the results for downstream analysis. The reference genome GRCh38.p14 was obtained from UCSC, and an integrated GTF annotation was generated by merging the GENCODE v48 annotation with all novel isoforms detected in the two cell lines after duplicate removal. The Y chromosome and patch chromosomes were excluded. AS events were defined using the generate Events module (-e SE SS MX RI FL) to produce ioe files with transcript and exon structures. AS event inclusion level (PSI) values were calculated at both event and isoform levels using psiPerEvent and psiPerIsoform. gDTUs were assessed with diffSplice (-m empirical -gc), providing ΔPSI values and empirical *p*-values. Only AS events detected in at least two out of three biological replicates were retained. Significant differential AS events and associated genes were defined by |ΔPSI| ≥ 0.1 and *p* < 0.05. Representative isoform structures and RNA-seq alignments were visualized using IGV 2.19.5 [40].

### 4.6. Differentially Expressed Genes and Differential Transcript Usage Analysis

Kallisto transcript-level quantifications were aggregated to gene-level counts using tximport (version 1.30.0), followed by differential gene expression analysis with DESeq2 (version 1.42.0), applying a significance threshold of *p* < 0.05. The gDTUs analysis was performed using the flair_diffSplice module of FLAIR, with default parameters plus the “--test” option. DRIMSeq was employed to detect significant splicing differences between conditions, retaining events with *p* < 0.05. Gene structure visualization was conducted using the IGV software (version 2.18.4). Both DEGs and gDTUs underwent GO and KEGG pathway enrichment analyses using the clusterProfiler package (version 4.1.4) in R. Adjusted *p*-values from the Benjamini–Hochberg method were used to assess enrichment significance.

## 5. Conclusions

Our study presents a transcriptome-wide comparison of 293T and A549 cells under GPCR overexpression using long-read sequencing. We identified DEGs and gDTUs across both native and receptor-overexpressing conditions by integrating FLAIR and SQANTI3 for comprehensive isoform-level analysis. By using this approach, we discovered novel isoforms, further enriching our understanding of cell-type-specific regulatory mechanisms. Subsequent experimental validation, particularly via PCR, will be necessary to advance functional studies on these newly discovered isoforms.

## Figures and Tables

**Figure 1 ijms-27-00487-f001:**
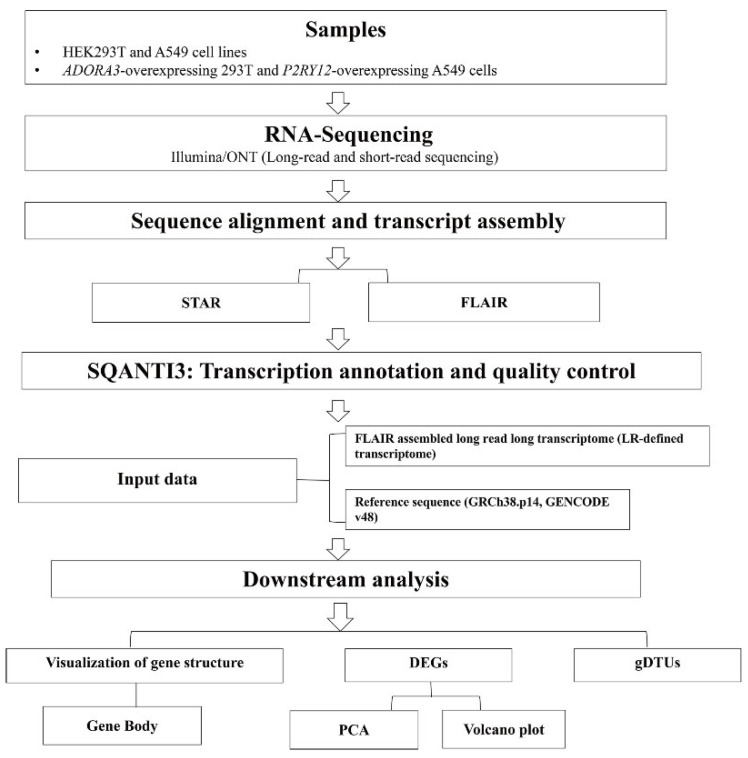
Integrated experimental and computational workflow (STAR: spliced transcripts’ alignment to a reference; FLAIR: full-length alternative isoform analysis of RNA; DEGs: differentially expressed genes; gDTUs: genes with differential transcript usages).

**Figure 2 ijms-27-00487-f002:**
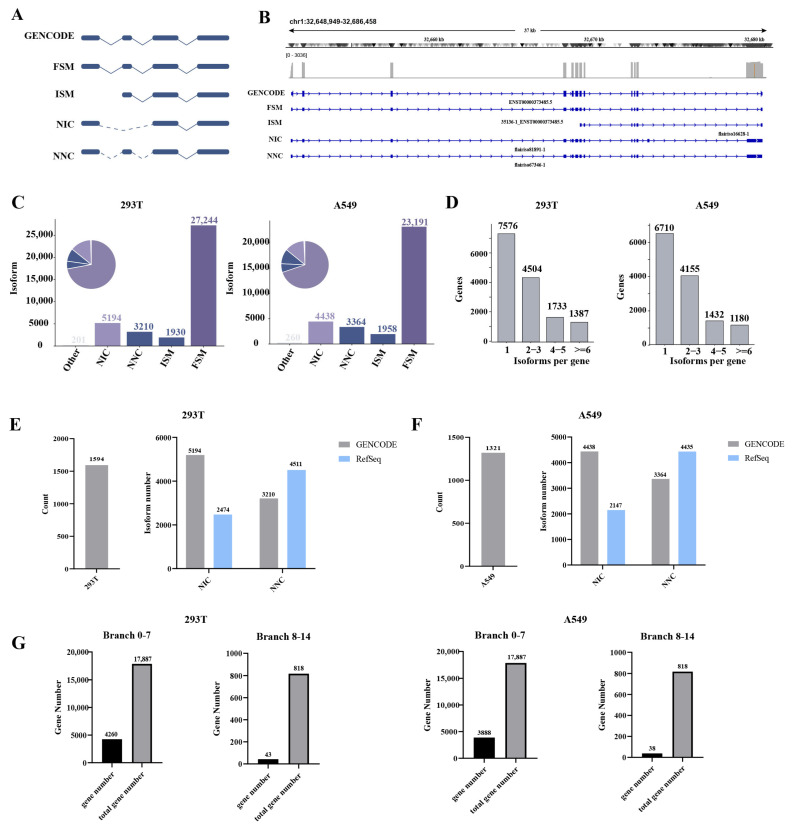
Long-read transcriptome analysis of 293T and A549 cells, leveraging SQANTI3 structural categories. (**A**) Schematics of SQANTI3, with defined transcript classes: full-splice match (FSM) (splicing patterns fully match known annotations); incomplete-splice match (ISM) (partial splicing matches); novel in catalog (NIC) (novel transcripts with partial overlap to known gene loci); and novel not in catalog (NNC) (completely novel transcripts with no homology to existing gene catalogs). These classify long-read transcript models per SQANTI3 criteria. (**B**) IGV-based visualization of a representative gene with all four SQANTI3 structural categories. A randomly selected gene (from SQANTI3 annotations) harboring FSM, ISM, NIC, and NNC structures is visualized in the Integrative Genomics Viewer (IGV). (**C**) Pie charts (proportions) and bar charts (absolute counts) quantify the distribution of SQANTI3-defined isoform categories in 293T and A549. FSM emerges as the most abundant class, while NIC/NNC contribute to novel transcript diversity, underscoring uncharacterized transcriptional output. (**D**) Bar graphs depict the number of genes with 1, 2–3, 4–5, or ≥6 isoforms in each cell line. (**E**–**G**) Comparative analysis of transcript isoform classification and gene age distribution between GENCODE and RefSeq annotations; cross-database comparison of isoform categories in (**E**) 293T and (**F**) A549 cells. (**G**) Evolutionary age distribution of genes with inconsistently annotated isoforms.

**Figure 3 ijms-27-00487-f003:**
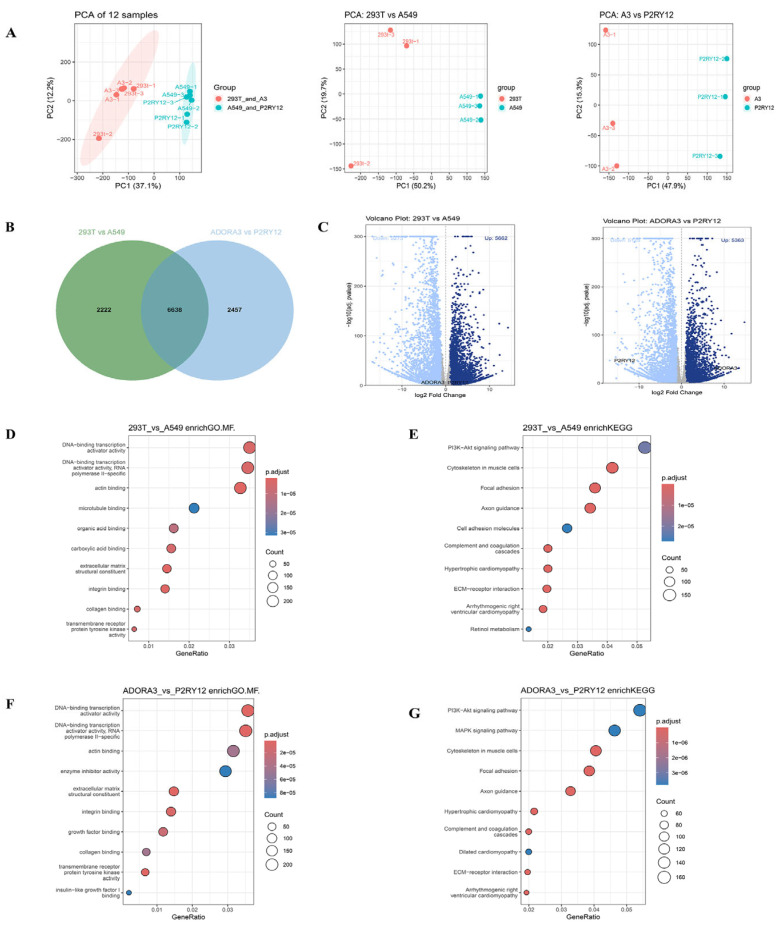
Transcriptome-wide comparison and functional characterization of 293T and A549 cells under baseline and receptor overexpression conditions. (**A**) Principal component analysis (PCA) of transcriptomes from 293T, A549, *ADORA3*-overexpressing 293T, and *P2RY12*-overexpressing A549 cells. (**B**) Venn diagram showing overlap of differentially expressed gene (DEG) sets between 293T vs. A549 and *ADORA3*-293T vs. *P2RY12*-A549 comparisons. (**C**) Bar plots displaying the number of upregulated and downregulated genes in 293T vs. A549 and *ADORA3*-293T vs. *P2RY12*-A549 comparisons. (**D**) Gene Ontology (GO) molecular function terms enriched in DEGs from 293T and A549 cells under baseline conditions. (**E**) KEGG pathway enrichment analysis of DEGs from 293T and A549 cells. (**F**) GO molecular function analysis following *ADORA3* and *P2RY12* overexpression. (**G**) KEGG pathway enrichment in *ADORA3*- and *P2RY12*-overexpressing cells. Red dots denote genes with up-regulated expression, and blue dots denote genes with down-regulated expression. Thresholds: padj < 0.05.

**Figure 4 ijms-27-00487-f004:**
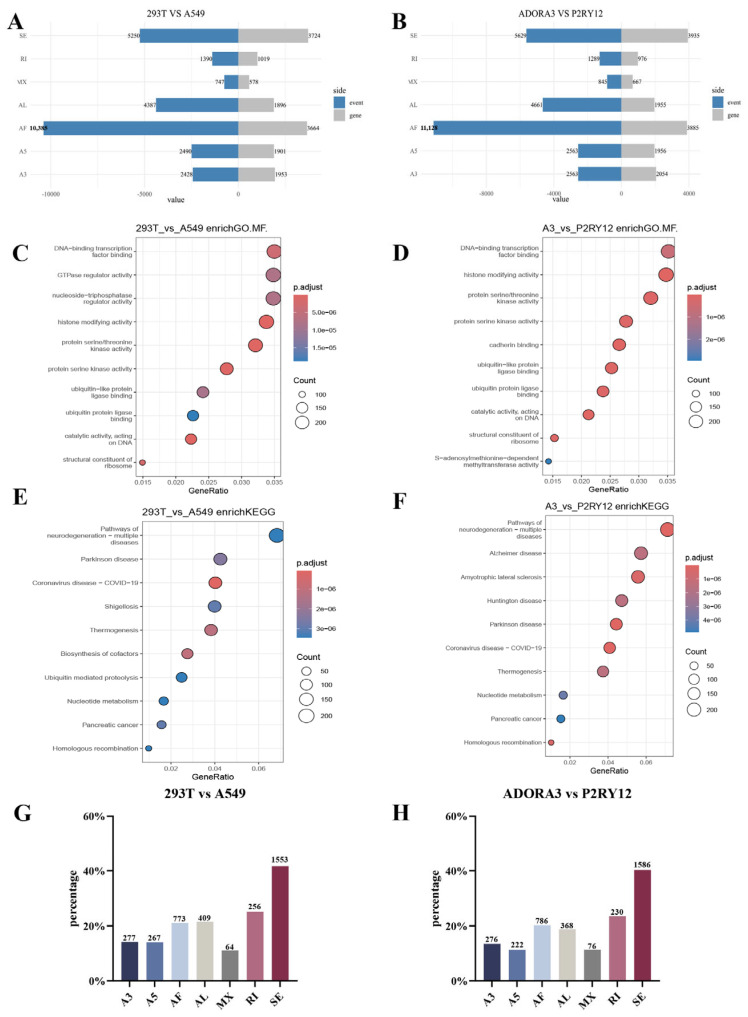
Functional enrichment analysis of gDTUs in 293T and A549 cells. (**A**) Distribution of alternative splicing (AS) events by type in 293T and A549 cells under baseline conditions. (**B**) Distribution of AS events in 293T cells overexpressing *ADORA3* and A549 cells overexpressing *P2RY12*. (**C**) Blue dots show genes with down-regulated expression, and red dots show genes with up-regulated expression; GO enrichment analysis of gDTUs between 293T and A549 cells under baseline conditions. (**D**) GO enrichment analysis of gDTUs in 293T cells overexpressing *ADORA3* and A549 cells overexpressing *P2RY12.* (**E**) KEGG pathway enrichment analysis of gDTUs in 293T and A549 cells. (**F**) KEGG pathway enrichment analysis of gDTUs in cells overexpressing *ADORA3* and *P2RY12*. (**G**) Overlap of gDEGs and gDTUs by splicing event type in baseline 293T and A549 cells. (**H**) Distribution of overlapping gDEG-gDTUs by splicing type in cells overexpressing GPCRs. (Gene ratio: The color gradient indicates different ranges of *p*.adjust values, reflecting the significance of enrichment. Count: The range of counts displayed is 50 to 200, representing the number of genes enriched in each pathway; larger circles indicate a higher number of enriched genes.).

**Figure 5 ijms-27-00487-f005:**
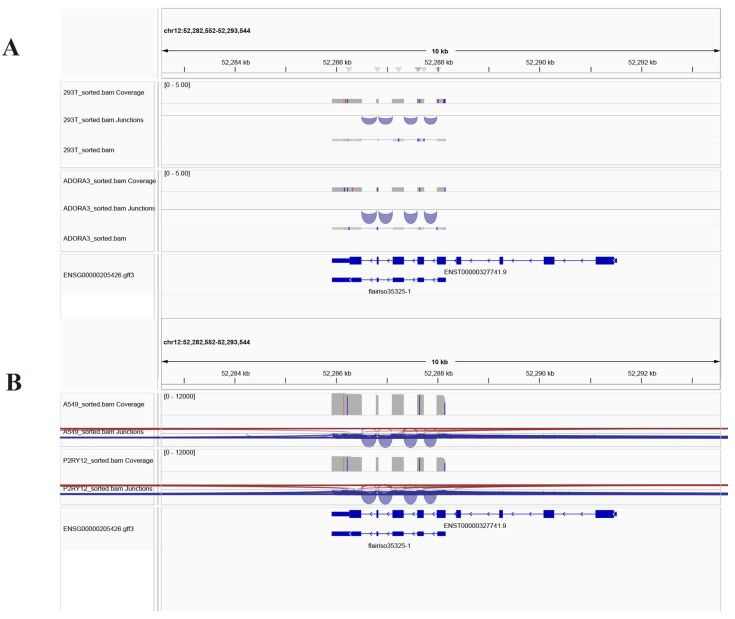
Comparative analysis of transcriptome alignment and splicing patterns. (**A**) IGV visualization of RNA-seq alignments at the KRT81 locus (chr12:52,282,552–52,293,544) in 293T and A549 cells; (**B**) IGV visualization of the same genomic region in A549 versus *P2RY12*-overexpressing A549 (*P2RY12*-1) cells. Red and blue represent splicing junctions on the positive and negative strands, respectively.

## Data Availability

The data presented in this study are openly available at the National Genomics Data Center at https://ngdc.cncb.ac.cn/bioproject/browse/PRJCA036438 (accessed on 21 February 2025; reference number: PRJCA036438) and at https://ngdc.cncb.ac.cn/search/all?q=PRJCA036439 (accessed on 21 February 2024; reference number: PRJCA036439).

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
