# Peer review of "Long-Read Sequencing Reveals Cell- and State-Specific Alternative Splicing in 293T and A549 Cell Transcriptomes"

_ijms, 2026, doi:10.3390/ijms27010487_

Round 1

Reviewer 1 Report

Comments and Suggestions for Authors

The manuscript of Li et al. “Long-Read Sequencing Reveals Cell- and State-Specific Alter-2 native Splicing in 293T and A549 Cell Transcriptomes” is devoted to the FLAIR- and SQANTI3-based analysis of full-length isoforms revealed by ONT long-read sequencing. Using such approach, the authors compared alternative splicing landscapes in 293T and A549 cultured cells under undisturbed and disturbed (overexpression of G protein-coupled receptors) conditions. They found that the overexpression of G protein-coupled receptors substantially changes the alternative splicing landscapes in both types of cells. The authors also revealed the presence of the remarkedly high number of novel transcript isoforms in 293T and A549 cultured cells. The manuscript is well-written, experiments are carefully designed, results are interesting and present in a clear and logic way. Yet, prior to accepting, a number of issues has to be addressed by the authors, which are outlined below.

  1. All figures are hard to read. The figure panels are too small. The designations of axis as well as designations on figure panels are unreadable. Please, rebuild the figures by enlarging the panels and using a larger size for fonts on figures to make them easy to read.
  2. Lines 96-98: Although some clarification for the isoform categories is given in the legend for Figure 2, it would be helpful to also provide a more detailed description of each category in the text, especially the last two.
  3. Lines 103-109: In fact, I do not see appreciable differences between 293T and A549 cells, based on the results present. I would recommend the authors to retract their claim that 293T cells have a more stable transcriptome and a greater potential for functional studies.
  4. Line 130: The results for 293T cells are given only in Figure 2E. Please, correct “Figures 2E-F” to “Figure 2E”.
  5. Lines 141-143: Please, rephrase to make it clear that the distribution of genes by age is analyzed but not the age distribution of genes between GENCODE and RefSeq.
  6. Lines 145-147: It would be helpful to provide more explanation on the meaning of branch 0-7 and branch 8-14 for reader's convenience.
  7. Figure 2G: The way the data are presented in Figure 2G may be misleading. The ordinate scale is logarithmic. While the number of genes is additive quantity, the logarithm of the gene number is not. I would recommend to present the data with separate histogram bars instead of two-colored histogram bars.
  8. Line 191: Correct the numbering of the figure. It must be Figure 3.
  9. Line 261: Please, correct “ADORA3 and P2RY12 cell lines” to “ADORA3-overexpressing 293T and P2RY12-overexpressing A549 cells”.
  10. The “Discussion” section (lines 444-471): Since the approach employed by the authors to treat the long-read and short-read sequencing data revealed the quite large number of novel transcript isoforms, I would recommend to add a pertinent discussion on possible reasons behind these findings.
  11. The “Conclusions” section (lines 560-566): A statement on a further requirement for PCR verification of novel transcript isoforms to validate the employed approach to a treatment of long-read sequencing data would be appropriate.

Author Response

The manuscript of Li et al. “Long-Read Sequencing Reveals Cell- and State-Specific Alter-2 native Splicing in 293T and A549 Cell Transcriptomes” is devoted to the FLAIR- and SQANTI3-based analysis of full-length isoforms revealed by ONT long-read sequencing. Using such approach, the authors compared alternative splicing landscapes in 293T and A549 cultured cells under undisturbed and disturbed (overexpression of G protein-coupled receptors) conditions. They found that the overexpression of G protein-coupled receptors substantially changes the alternative splicing landscapes in both types of cells. The authors also revealed the presence of the remarkedly high number of novel transcript isoforms in 293T and A549 cultured cells. The manuscript is well-written, experiments are carefully designed, results are interesting and present in a clear and logic way. Yet, prior to accepting, a number of issues has to be addressed by the authors, which are outlined below.

Thank you very much for dedicating your time and expertise to review our manuscript, "Long-Read Sequencing Reveals Cell- and State-Specific Alter-native Splicing in 293T and A549 Cell Transcriptomes", and providing detailed and constructive feedback. We deeply appreciate the opportunity to enhance our work based on your insights, which have significantly improved the manuscript's quality, scientific rigor, and precision.

Comment 1: All figures are hard to read. The figure panels are too small. The designations of axis as well as designations on figure panels are unreadable. Please, rebuild the figures by enlarging the panels and using a larger size for fonts on figures to make them easy to read.

Respond 1:

Thank you for your valuable feedback regarding the readability of the figures in our manuscript. We have carefully reviewed all figures and agree with your assessment. In accordance with your suggestion, we have thoroughly revised all figures to improve their clarity. Specifically, we have: enlarged all figure panels to maximize visibility, significantly increased the font size for all axis labels, legends, and data point designations, and adjusted the layout and contrast to ensure all graphical elements are clear and easy to read. These modifications have been incorporated into the revised version of the manuscript. We hope the updated figures now meet the necessary standards for presentation and readability.

Comment 2: Lines 96-98: Although some clarification for the isoform categories is given in the legend for Figure 2, it would be helpful to also provide a more detailed description of each category in the text, especially the last two.

Respond 2:

Thank you for the valuable suggestion. We agree that providing a more detailed description of the classification categories in the main text would be helpful for readers' understanding. Following the your suggestion, we added detailed explanations of the SQANTI3 classification categories in the main text.

Lines 96-107

We classified all high-confidence transcript isoforms identified using the SQANTI3 tool based on the GENCODE v48 reference annotation. The specific categories are as follows: Full-Spliced Match (FSM), which refers to isoforms that exactly match all splice junctions of an annotated reference transcript; Incomplete-Spliced Match (ISM), where isoforms align with a reference transcript's splice junctions but show truncations or extensions at either the 5' or 3' end; Novel in Catalog (NIC), denoting isoforms that are new combinations of known, annotated splice junctions; and Novel Not in Catalog (NNC), consisting of isoforms containing at least one new splice junction not present in the reference annotation. Additionally, antisense, fusion, genic, genic intron, and intragenic transcripts were grouped into the 'other' category (Figure 2A).

Through this addition, we clearly distinguish the key differences between NIC (new combinations of known elements) and NNC (containing novel splice elements), which helps readers to understand the types of novel transcripts identified in our analysis.

Comment 3: Lines 103-109: In fact, I do not see appreciable differences between 293T and A549 cells, based on the results present. I would recommend the authors to retract their claim that 293T cells have a more stable transcriptome and a greater potential for functional studies.

Respond 3:

Thank you very much for your valuable feedback. We fully agree with your perspective. Upon re-evaluation, the absolute difference in transcriptomic annotation concordance between the two cell lines is indeed limited (e.g., only about a 1.5% difference in the match rate). These data are insufficient to support the inferential conclusion that "293T cells have a more stable transcriptome and greater potential for functional studies." Interpreting this minor difference as indicative of a more stable transcriptome does indeed appear unwarranted.

Based on your suggestion, we removed this problematic conclusion regarding transcriptome stability and potential for functional studies from lines 103-109.

Comment 4: Line 130: The results for 293T cells are given only in Figure 2E. Please, correct “Figures 2E-F” to “Figure 2E”.

Respond 4:

Thank you for carefully reviewing the manuscript and pointing out this error. We have corrected "Figures 2E-F" to "Figure 2E" in line 130 of the original text.

Comment 5: Lines 141-143: Please, rephrase to make it clear that the distribution of genes by age is analyzed but not the age distribution of genes between GENCODE and RefSeq.

Respond 5:

Thank you for the insightful comment. We agree that the original phrasing may create ambiguity regarding what distributions were compared. We intended to analyze the evolutionary age distribution of genes that harbor inconsistently annotated isoforms between GENCODE and RefSeq, not to compare the age distributions of genes between the two annotation databases.

To clarify this point, we have revised the sentence in the Methods/Results section as follows:

Lines 149-151

To investigate how gene evolutionary age influences annotation inconsistency, we analyzed the gene age distribution of protein-coding genes that harbor isoforms inconsistently annotated between GENCODE and RefSeq in 293T and A549 cells.

Comment 6: Lines 145-147: It would be helpful to provide more explanation on the meaning of branch 0-7 and branch 8-14 for reader's convenience.

Respond 6:

Thank you for the helpful suggestion. Although our text mentioned that branches 0–7 correspond to evolutionarily ancient, non-primate-specific genes, we agree that the meaning of the branch classification should be further clarified. Accordingly, we have added the following explanation in the revised manuscript:

“In this gene-age classification system, branches 0–7 represent genes that originated prior to the primate lineage, whereas branches 8–14 correspond to primate-specific genes.”

This clarification is now included immediately after the description of Figure 2G.

we have revised the sentence in the Results section as follows:

Lines 151-157

As shown in Figure 2G, the vast majority of these genes were evolutionarily ancient, non-primate-specific genes (branch 0–7). In this context, in 293T cells, 4,260 out of 17,887 total genes in branch 0–7 contained such isoforms, compared to only 43 out of 818 primate-specific genes (branch 8–14). In this gene-age classification system, branches 0–7 represent genes that originated prior to the primate lineage, whereas branches 8–14 correspond to primate-specific genes.

Comment 7: Figure 2G: The way the data are presented in Figure 2G may be misleading. The ordinate scale is logarithmic. While the number of genes is additive quantity, the logarithm of the gene number is not. I would recommend to present the data with separate histogram bars instead of two-colored histogram bars.

Respond 7:

Thank you for this insightful comment regarding the presentation of data in Figure 2G. We agree with your perspective and have revised Figure 2G as recommended, redrawing it as separate histogram bars to present the data more clearly.

Comment 8: Line 191: Correct the numbering of the figure. It must be Figure 3.

Respond 8:

Thank you for pointing out this mistake. We have updated the text at line 191 as suggested. The figure reference has been corrected to "Figure 3" in the revised manuscript.

Comment 9: Line 261: Please, correct “ADORA3 and P2RY12 cell lines” to “ADORA3-overexpressing 293T and P2RY12-overexpressing A549 cells”.

Respond 9:

Thank you for your careful correction. Following your suggestion, we have revised the phrase "ADORA3 and P2RY12 cell lines" in the manuscript to "ADORA3-overexpressing 293T and P2RY12-overexpressing A549 cells" in Lines 270271.

Comment 10: The “Discussion” section (lines 444-471): Since the approach employed by the authors to treat the long-read and short-read sequencing data revealed the quite large number of novel transcript isoforms, I would recommend to add a pertinent discussion on possible reasons behind these findings.

Respond 10:

Thank you for this valuable suggestion. We added a paragraph to discuss the identification of novel isoform in the disccusion part:

Line 412-430

Alternative splicing has maintained its significance in biological research since the discovery of introns, primarily due to its pivotal role in fundamental processes like gene expression regulation (Gilbert Walter, 1978; Keren Hadas et al., 2010). Nevertheless, research in this area is hampered by the fact that the similarity between isoforms from the same gene can be confused with other factors, such as similarities among genes in the same family or other repetitive regions within genes (Kanitz 2015). Our prior research has demonstrated the viability of combining second- and third-generation sequencing methodologies for identifying novel plant isoforms, facilitated by the advent of third-generation sequencing in isoform characterization (Ma et al., 2021; Gao et al., 2019; Liu et al., 2023). Similarly, third-generation sequencing has enabled the discovery of novel isoforms across diverse human tissues and cell types (Zhu et al., 2021, Su et al., 2024, Inamo et al., 2024). The identification of novel isoforms and gDTUs continues to face certain limitations. The utilization of only three replicates in this study, a consequence of resource restrictions, may exert an influence on the reliability of alternative splicing analysis, primarily because this analysis is acutely vulnerable to both biological noise and experimental batch effects. Furthermore, the inherent variability observed in ONT reads represents an often-overlooked aspect in the experimental design, a challenge that could be addressed through techniques like spike-in RNA quantification (Byrne et al., 2017; Glinos 2022). Thus, it is imperative to note that while these newly identified isoforms are supported by RNA-seq data, they necessitate additional experimental validation, such as PCR verification, prior to proceeding with functional investigations.

Comment 11: The “Conclusions” section (lines 560-566): A statement on a further requirement for PCR verification of novel transcript isoforms to validate the employed approach to a treatment of long-read sequencing data would be appropriate.

Respond 11:

Thank you for your suggestion. We revised the discussion and conclusion sections to highlight the crucial need for experimental validation preceding functional studies of novel isoforms.

Line 578-579

Subsequent experimental validation, particularly via PCR, is a prerequisite for advancing functional studies on these newly discovered isoforms.

Reviewer 2 Report

Comments and Suggestions for Authors

This work focuses on identifying alternative splicing variants via long-read sequencing.

The work does not present results on the biological role of messenger RNA variant diversity in two cell models. It focuses on a descriptive and technical analysis, which may be of interest to those working with this sequencing tool and on alternative splicing analysis; however, I do not believe this journal is the appropriate venue for this type of work.

Suggestions

Methodology

  1. Although the data have already been published (https://ngdc.cncb.ac.cn/bioproject/browse/PRJCA036438) (https://ngdc.cncb.ac.cn/search/all?q=PRJCA036439), the sequencing results are not accessible. If the results are already available, please include the experiment IDs. Lines 479-480.

  1. It is necessary to add information about the sequencing platform used, the protocol, the number of reads obtained, quality control, etc., even though the data are already available.

Results.

  1. Lines 114-119. The results as presented are unclear. In the 293T analysis, the percentage of transcripts with more than one variant was mentioned (90.24% FSM, 78.09% FSM, and 67.66% FSM). However, the data presented do not represent 100% of the identified transcripts. The results were similar for the A549 cell line.

  1. The figures in the manuscript are of poor quality, unlike the downloadable figures.

  1. In Figure 1 gDTUs, the number of readings cannot be discerned; the letters and numbers overlap. Although they provide a description and classification of the identified isoforms, they do not offer biological significance for the results.

Author Response

Thank you for your careful review of our manuscript and for providing valuable feedback. This investigation is primarily a descriptive and technical analysis. Earlier, we conducted comparisons of transcriptomic variations between 293T and ADORA3-overexpressing 293T cells, along with A549 and P2RY12-overexpressing A549 cells. Our main objective in those studies was to understand how the overexpression of a particular gene could alter the transcriptome of 293T and A549 cell lines, specifically concerning differential expression and splicing alternation (PMID: 40362672 and PMID: 40243586). In this current work, the FLAIR-SQUANTI3 pipeline was employed to pinpoint novel isoforms within these two cell lines. A feasible strategy for examining transcriptomic modifications within cell lines, specifically alternative splicing, was established through these analyses. In our manuscript titled "Long-Read Sequencing Reveals Cell- and State-Specific Alternative Splicing in 293T and A549 Cell Transcriptomes," we would like to clarify the following points and commit to making significant revisions to enhance the value of the work.

Comment 1: Although the data have already been published (https://ngdc.cncb.ac.cn/bioproject/browse/PRJCA036438) (https://ngdc.cncb.ac.cn/search/all?q=PRJCA036439), the sequencing results are not accessible. If the results are already available, please include the experiment IDs. Lines 479-480.

Respond 1:

Thank you for your careful review and for pointing this out. We have rechecked all data accessibility information and updated the manuscript with the correct experiment-level access links and IDs.

Our study includes two categories of sequencing data:

  1. Previously released short-read RNA-seq datasets

For PRJCA036438, the sequencing reads and detailed experimental information are available through the GSA-Human page HRA010505: https://ngdc.cncb.ac.cn/gsa-human/browse/HRA010505

For PRJCA036439, sequencing reads and experimental information can be accessed through HRA010506: https://ngdc.cncb.ac.cn/gsa-human/browse/HRA010506

The earlier BioProject pages contained multiple sub-links, which may have made it difficult to locate the specific download pages. We have now replaced them with direct access links to the experiment-level GSA-Human pages.

  1. Oxford Nanopore long-read sequencing data generated in this study

These data have been submitted to the NGDC under BioProject PRJCA052561. sequencing reads, and experimental information can be accessed through HRA015092: https://ngdc.cncb.ac.cn/gsa-human/browse/HRA015092. After the human genetic resource information of this dataset is registered, the download link for the source data can be released.

The revised manuscript now includes all accessible links, BioProject numbers, and experiment IDs to ensure complete transparency and reproducibility.

we have revised the sentence in the Methods section as follows:

Lines 459-469

All sequencing data used in this study are publicly available at the National Genomics Data Center (NGDC). The short-read RNA-seq datasets previously published [8-9] under BioProjects PRJCA036438 and PRJCA036439 can be accessed at the GSA-Human experiment-level pages HRA010505 (https://ngdc.cncb.ac.cn/gsa-human/browse/HRA010505) and HRA010506 (https://ngdc.cncb.ac.cn/gsa-human/browse/HRA010506), respectively, where the sequencing reads and detailed experimental information are provided. The Oxford Nanopore long-read sequencing data generated in this study have been submitted under BioProject PRJCA052561, and the corresponding submission information is available at the GSA-Human page subHRA023356 (https://ngdc.cncb.ac.cn/gsa-human/submit/hra/subHRA023356). All access links and experiment IDs have been updated to ensure full transparency and reproducibility.

Comment 2: It is necessary to add information about the sequencing platform used, the protocol, the number of reads obtained, quality control, etc., even though the data are already available.

Respond 2:

Thank you for your valuable comments. In response, we have added a dedicated paragraph in the Materials and Methods section to more clearly describe both the long-read (Nanopore) and short-read (Illumina RNA-seq) sequencing workflows. For the short-read RNA-seq data, we now provide a concise summary of the sequencing strategy for the four experimental groups—293T, ADORA3-overexpression, A549, and P2RY12-overexpression—comprising a total of 12 libraries. Key quality-control metrics, including read counts and base quality scores, have also been reported. These revisions enhance the transparency of our sequencing procedures and data quality assessment.

We have added a paragraph in the Methods section as follows:

Lines 470-485

Sequencing and Quality Control

For short-read RNA-seq, Illumina libraries were generated for four groups: 293T, ADORA3-overexpression, A549, and P2RY12-overexpression (n = 3 biological replicates per group). Total RNA was extracted using TRIzol, followed by rRNA depletion and library construction using the MGIEasy Circularization Kit (Cat# 1000004155). Sequencing was performed on an Illumina platform (DNBSEQ-T7) to obtain 150-bp paired-end reads. Quality assessment was conducted using FastQC (v0.12.1) and Trimmomatic(v.0.39), with all samples showing high base accuracy (Q30 > 93%) and adequate sequencing depth for differential expression analysis. These short-read data served to complement the long-read quantification and to support the validation of differential expression findings. For long-read transcriptome profiling, sequencing was performed on the PromethION platform following the manufacturer’s protocol. Basecalling was carried out with Guppy, and low-quality Nanopore reads (mean Phred quality ≤ 7) were removed before downstream analysis. Across the 12 libraries, we obtained approximately 4.4–6.5 million long reads per sample, with N50 values ranging from 1.3 to 1.7 kb and mean Q scores of 16–17 after filtering.

Comment 3: Lines 114-119. The results as presented are unclear. In the 293T analysis, the percentage of transcripts with more than one variant was mentioned (90.24% FSM, 78.09% FSM, and 67.66% FSM). However, the data presented do not represent 100% of the identified transcripts. The results were similar for the A549 cell line.

Respond 3:

We sincerely appreciate your valuable feedback regarding the clarity of the results presented in lines 114–119 of our manuscript. We fully agree that the original description of the percentage of transcripts containing more than one variant was not sufficiently clear and did not adequately convey their distribution across all identified transcripts.

In response to your suggestion, we have revised and clarified the relevant descriptions in lines 119–132 of the revised manuscript.

Lines 118–131

To further examine transcript diversity across genes, we assessed isoform counts per gene. In 293T cells, 6,710 genes expressed a single isoform, 4,504 genes expressed 2–3 isoforms, 1,733 genes expressed 4–5 isoforms, and 1,387 genes expressed ≥6 isoforms. A549 cells showed similar trends: 6,709 genes with one isoform, 4,155 genes with 2–3 isoforms, 1,432 genes with 4–5 isoforms, and 1,178 genes with ≥6 isoforms (Figure 2D). Besides, 293T cells harbored more genes with ≥6 isoforms (1,387 vs. 1,178). This observed difference, though noted within broadly similar overall distributions, is consistent with a somewhat higher transcriptional complexity in 293T cells. In both lines, isoform counts per gene approximately follow a long-tail distribution, with much of the transcript diversity concentrated in a limited subset of genes that produce numerous isoforms. The variation in high-complexity genes might reflect cell-type-specific differences in splicing regulation, which could be linked to distinct developmental, functional, or pathological contexts. Together, these observations provide a basis for considering how transcript diversity may be organized and regulated across different cell types.

The main modifications are as follows: We hope that with the above revisions, the presentation of the relevant results is now clearer and more rigorous. Once again, we sincerely appreciate your thorough and professional review, which has been greatly valuable in improving the quality of our manuscript.

Comment 4: The figures in the manuscript are of poor quality, unlike the downloadable figures.

Respond 4:

Thank you for highlighting this important point regarding figure quality. We agree that the resolution and clarity of figures are critical for accurate interpretation. In the initial submission, the figures embedded within the main manuscript text may have been compressed to meet file size requirements, which likely resulted in the perceived quality difference compared to the higher-resolution downloadable files.

Comment 5: In Figure 1 gDTUs, the number of readings cannot be discerned; the letters and numbers overlap. Although they provide a description and classification of the identified isoforms, they do not offer biological significance for the results.

Respond 5:

We sincerely appreciate your valuable feedback regarding both the graphical presentation in Figure 1 and the interpretation of our results. We fully agree with your suggestions and have undertaken significant revisions to address them.

We have redesigned Figure 1 into a plain format. This new presentation more clearly summarizes the key comparative results and their implications regarding cell-type-specific splicing responses.

Reviewer 3 Report

Comments and Suggestions for Authors

Summary of the Manuscript

The manuscript presents an integrative long-read/short-read transcriptomic analysis of HEK293T and A549 cells under both basal conditions and after GPCR overexpression (ADORA3 in 293T; P2RY12 in A549). The authors use ONT long-read sequencing combined with FLAIR, SQANTI3, SUPPA2, DESeq2, and DRIMSeq to characterize isoform diversity, alternative splicing events, differentially expressed genes, and differential transcript usage.

Key findings include:

  • Both cell lines express large numbers of novel isoforms (18–20%).
  • 293T cells show higher annotation concordance (FSM >80%) and fewer novel isoforms.
  • GPCR overexpression induces widespread transcriptomic remodeling.
  • A549 exhibits stronger transcriptional plasticity, consistent with oncogenic behavior.
  • Exon skipping (SE) and alternative first/last exons dominate AS patterns.
  • KEGG/GO analysis reveals differences in metabolic, signaling, and splicing-related pathways.
  • The authors conclude that 293T is well suited for splicing-regulation studies, while A549 is more suited for tumor-associated transcriptome investigations.

MAJOR ISSUES

These points concern scientific rigor, experimental design, analysis, and interpretation. They must be addressed for the manuscript to meet IJMS standards.

  1. Study Design Limitations Not Fully Acknowledged

The study uses:

n = 3 biological replicates per condition

GPCR overexpression, which itself creates strong artificial transcriptomic effects

No independent validation datasets, rescue experiments, or wet-lab validation

Because alternative splicing analysis is highly sensitive to biological noise and batch effects, n=3 is minimal for long-read studies.

Required revisions:

Add a limitations subsection explicitly stating the restricted sample size, lack of functional validation, and potential variability in ONT reads.

Clarify how batch effects were controlled (batching of 293T and A549 sequencing not shown).

  1. Over-interpretation of Pathway and GO Results

Much of the KEGG/GO analysis includes:

Very broad, non-specific pathways (e.g., “Human cytomegalovirus infection”, “Proteoglycans in cancer”, “Neurodegeneration pathways”)

Pathways unlikely to be directly relevant to overexpressed GPCRs or baseline splicing differences

Very long lists (hundreds of genes), making interpretation noisy

These functional analyses, as written, read as gene-set inflation rather than targeted mechanistic insight.

Required revisions:

Drastically reduce the number of pathways reported—highlight the most biologically relevant ones only.

Avoid causal language unless directly tested.

Emphasize exploratory nature and avoid overstating disease relevance.

  1. Novel Isoform Identification Requires Stronger Quality Control Justification

The manuscript reports:

10,000 non-FSM isoforms

~3,000 NNC transcripts

Large annotation discrepancies between GENCODE and RefSeq (Figure 2E–F)

However:

No false positive estimation is presented.

Novel isoform validation is not shown (junction coverage, canonical splice site distribution, read count thresholds).

Required revisions:

Provide summary QC metrics: splice site canonicality, read depth distributions.

Clarify the minimum read support thresholds used by SQANTI3.

Add examples in supplementary figures showing raw ONT read support for selected novel isoforms.

  1. GPCR Overexpression Confounds Interpretation

The manuscript aims to analyze splicing changes between cell types, but:

The perturbation models differ in both receptor and cell type (ADORA3 in 293T vs P2RY12 in A549).

It is impossible to disentangle cell-type effects from GPCR-specific effects.

Required revisions:

Add explicit text explaining that interpretations of differences between overexpression groups cannot separate receptor-driven vs. cell-line-driven effects.

Tone down conclusions attributing changes directly to receptor signaling.

  1. Absence of Short-Read Validation of DTU / AS Events

Although short-reads are used for FLAIR correction, they are not used to validate PSI values or confirm AS events.

Required revisions:

Validate at least 3–5 key AS events using short-read junction counts (e.g., sashimi plots).

Alternatively, explicitly state this limitation and mark the results as exploratory.

  1. Figures Require Better Labeling and Clearer Interpretation

Examples:

Figure 2B: IGV screenshot lacks scale, track labels, and read depth.

Figures 3E–H & 4C–F: Dot plots are densely packed and difficult to interpret without color legends.

Figure 5: Novel isoform detection at KRT81 locus shown, but no quantitative read coverage or evidence.

Required revisions:

Improve figure resolution and readability.

Add clear legends, axis labels, and annotation tracks.

Provide more representative examples of splicing differences.

  1. Discussion Section Is Descriptive Rather Than Analytical

The discussion mostly restates results. It lacks:

Mechanistic hypotheses

Integration with relevant literature on GPCR-induced splicing regulation

Explanation of why A549 is more plastic beyond “it is a cancer cell line”

Required revisions:

Add mechanistic interpretation tied to known splicing regulators.

Provide context: how do these findings compare to prior long-read datasets?

Expand on biological implications of alternative first/last exons (AF/AL), which are unexpectedly abundant.

MINOR ISSUES

  1. English grammar, clarity, and conciseness

Several sections require language polishing for clarity (e.g., introduction and results are lengthy and repetitive).

  1. Add a graphical abstract

Given the multi-dimensional analysis, a graphical summary would help readers.

  1. Methods need more explicit parameter reporting

Examples: FLAIR parameters, SUPPA2 ΔPSI threshold justification, SQANTI3 rescue criteria.

  1. PCA plots (Figure 3A) require confidence ellipses or sample labels

Currently, it is unclear if replicates cluster tightly.

  1. Ensure consistent terminology

“gDTU” vs “DTU genes” vs “AS events” interchanged inconsistently.

  1. Tone down statements implying broadly applicable biological conclusions

e.g., “These findings reflect tumor-specific processes” should be “These findings may reflect…”

  1. Citations need minor corrections

Some references are missing publication details or have formatting inconsistencies.

Author Response

Thank you very much for investing your valuable time and effort in reviewing our manuscript and providing such valuable and professional comments. The limitations in the study design that you pointed out are of great importance, and we have carefully considered them and made corresponding additions and clarifications in the text. Your comments have significantly enhanced the rigor and transparency of our study. We hope that the additions and revisions we have made effectively address your concerns. Once again, we sincerely appreciate your insightful review and assistance with our work.

Comment 1: Study Design Limitations Not Fully Acknowledged

The study uses:

n = 3 biological replicates per condition

GPCR overexpression, which itself creates strong artificial transcriptomic effects. No independent validation datasets, rescue experiments, or wet-lab validation. Because alternative splicing analysis is highly sensitive to biological noise and batch effects, n=3 is minimal for long-read studies.

Required revisions:

Add a limitations subsection explicitly stating the restricted sample size, lack of functional validation, and potential variability in ONT reads. Clarify how batch effects were controlled (batching of 293T and A549 sequencing not shown).

Respond 1:

We appreciate your insightful feedback concerning the limitations of this study. Consequently, we have incorporated a section addressing these limitations within the manuscript:

Lines 421-430

The identification of novel isoforms and gDTUs continues to face certain limitations. The utilization of only three replicates in this study, a consequence of resource restrictions, may exert an influence on the reliability of alternative splicing analysis, primarily because this analysis is acutely vulnerable to both biological noise and experimental batch effects. Furthermore, the inherent variability observed in ONT reads represents an often-overlooked aspect in the experimental design, a challenge that could be addressed through techniques like spike-in RNA quantification (Byrne et al., 2017; Glinos, 2022). Thus, it is imperative to note that while these newly identified isoforms are supported by RNA-seq data, they necessitate additional experimental validation, such as PCR verification, prior to proceeding with functional investigations.

Building upon prior investigations (PMID: 40362672 and PMID: 40243586), which analyzed transcriptomic variations between 293T and ADORA3‑overexpressing 293T cells, as well as A549 and P2RY12‑overexpressing A549 cells, this study sought to elucidate the transcriptomic disparities between 293T and A549 cells and to identify novel isoforms within these two cell lines. Figure 1-A was subsequently adjusted to enable the comparative assessment of all twelve samples.

Figure 1-A

Comment 2: Over-interpretation of Pathway and GO Results

Much of the KEGG/GO analysis includes:

Very broad, non-specific pathways (e.g., “Human cytomegalovirus infection”, “Proteoglycans in cancer”, “Neurodegeneration pathways”). Pathways unlikely to be directly relevant to overexpressed GPCRs or baseline splicing differences. Very long lists (hundreds of genes), making interpretation noisy. These functional analyses, as written, read as gene-set inflation rather than targeted mechanistic insight.

Required revisions:

Drastically reduce the number of pathways reported—highlight the most biologically relevant ones only. Avoid causal language unless directly tested. Emphasize exploratory nature and avoid overstating disease relevance.

Respond 2:

Thank you for this critical and constructive comment. We sincerely appreciate the reviewer's insight, as it helped us recognize that the initial functional analysis was overly broad and lacked focus. We have thoroughly revised this section to address your concerns: To enhance the rigor and focus of the analysis, the functional enrichment analysis tool was switched from DAVID to the R package clusterProfiler, which directly calls locally and continuously updated species-specific annotation packages (e.g.,org.Hs.eg.db) to ensure the accuracy of the analytical foundation. The pathway lists have been significantly refined and filtered to highlight only the most pertinent and biologically relevant terms.

  • Figures 4C-F

We have carefully reviewed the text and replaced causal language with associative or observational phrasing to ensure accuracy.

The narrative now explicitly underscores the exploratory nature of this analysis, and we have tempered the discussion to avoid overinterpretation.

These modifications, which reflect your valuable suggestions, can be found in the revised manuscript on Lines 299-322.  

GO enrichment analysis identified distinct molecular function profiles in gDTUs between 293T and A549 cells (Figure 4C) and between ADORA3 and P2RY12-related cells (Figure 4D). In the 293T vs. A549 comparison, significant enrichments were observed for DNA-binding transcription factor activity, GTPase regulator activity, histone-modifying activity, and protein serine/threonine kinase activity (adjusted p-value < 0.05). These patterns likely reflect differences in transcriptional regulation, signal transduction, and epigenetic modification between the cell lines. Additionally, enrichment of genes related to ubiquitin-like protein ligase binding may indicate variations in protein stability regulation.

In the ADORA3 vs. P2RY12 comparison, notable enrichments included protein serine/threonine kinase activity, protein serine kinase activity, and catalytic activity acting on DNA. These findings suggest potential differences in gene expression regulation, signal transduction, protein homeostasis, and epigenetic modification between the two cell types. Overall, these results provide insights into the molecular distinctions between the cell lines and highlight areas for further exploration of their biological mechanisms.

Figures 4E and 4F show the KEGG pathway enrichment analysis results for 293T vs. A549 cells and ADORA3 vs. P2RY12-related cells, respectively. These analyses identified both differences and commonalities in several key biological pathways across the cell lines. For instance, the enriched pathways in both comparisons included neurodegenerative diseases (e.g., Alzheimer's and Parkinson's disease), COVID-19, thermogenesis, nucleotide metabolism, and, in the case of 293T vs. A549, ubiquitin-mediated proteolysis. These findings may reflect potential variations in how the cell lines handle neurodegenerative mechanisms, energy metabolism, DNA repair, and responses to viral infections. Overall, these results provide useful insights that could inform further studies into the functional distinctions and underlying biological mechanisms between these cell types.

We hope these changes have improved the clarity and appropriateness of our interpretation.

Comment 3: Novel Isoform Identification Requires Stronger Quality Control Justification

The manuscript reports:

10,000 non-FSM isoforms ~3,000 NNC transcripts. Large annotation discrepancies between GENCODE and RefSeq (Figure 2E–F)

However:

No false positive estimation is presented. Novel isoform validation is not shown (junction coverage, canonical splice site distribution, read count thresholds).

Required revisions:

Provide summary QC metrics: splice site canonicality, read depth distributions.

Clarify the minimum read support thresholds used by SQANTI3.

Add examples in supplementary figures showing raw ONT read support for selected novel isoforms.

Respond 3:

Thank you for the critical feedback regarding the validation of novel isoforms. We are pleased to provide the following detailed quality control (QC) information, which has been incorporated into the revised manuscript and Supplementary Materials.

  1. Summary QC Metrics for Novel Isoforms

A comprehensive summary of QC metrics for all identified novel isoforms (both NIC and NNC) is now provided.

Splice Site Canonicality: Analysis of splice junctions within novel isoforms revealed a very high rate of canonical splice sites. For the 293T cell line, 99.14% of NIC and 99.09% of NNC junctions were canonical (GT-AG, GC-AG, or AT-AC). This pattern was consistent in A549 cells. NIC would involve new junctions, whereas all junctions within NNC are previously documented in GENCODE or refSeq annotations. However, our findings revealed similar proportions of canonical and non-canonical splice sites across both NIC and NNC groups.

Supplementary 1. The number of splice site in 293T and A549 cell lines. GT-AT, GC-AG, and AT-AC are considered canonical types; all remaining types are consequently non-canonical.

Read Support (Junction Coverage): Isoform-level reliability was assessed using the min_sample_cov metric from SQANTI3, which reports the minimum number of samples providing short-read evidence supporting all junctions of an isoform. The distribution of this metric for NIC/NNC isoforms is now provided. Importantly, downstream differential analysis was restricted to isoforms with robust support (min_sample_cov ≥ 1 in at least one experimental group).

  1. Clarification of SQANTI3 Parameters and Support Thresholds

We have clarified the analysis pipeline and parameters in the Methods section.

SQANTI3 Quality Control Mode: SQANTI3 (v5.4) was run in QC mode without applying its filter/rescue module to preserve isoform diversity. It annotated isoforms using short-read alignments (STAR) for junction/TSS/TES support, CAGE peaks, and polyA motifs (with a 50 bp window for TTS).

Read Support Threshold: SQANTI3 QC does not inherently filter isoforms by read count. Initial isoform calling with FLAIR collapse used a default minimum of 1 full-length ONT read. The key filtering for downstream analysis was based on SQANTI3's min_sample_cov (short-read junction support across replicates) as described above, ensuring isoforms were supported by orthogonal evidence.

  1. Evidence of Raw ONT Read Support

In accordance with the suggestion, Figure 5 has been modified to present the raw ONT read alignments. These alignments were subsequently collapsed within the "*.bam Coverage" tracks. Consequently, every read visualized in these tracks represents a raw ONT read.

In summary, the revised manuscript includes expanded QC statistics, clarified methodological thresholds, and visual validation, which collectively strengthen the justification for the novel isoforms reported.

Comment 4: GPCR Overexpression Confounds Interpretation

The manuscript aims to analyze splicing changes between cell types, but:

The perturbation models differ in both receptor and cell type (ADORA3 in 293T vs P2RY12 in A549). It is impossible to disentangle cell-type effects from GPCR-specific effects.

Required revisions:

Add explicit text explaining that interpretations of differences between overexpression groups

cannot separate receptor-driven vs. cell-line-driven effects.

Tone down conclusions attributing changes directly to receptor signaling.

Respond 4:

Thank you for this critical comment regarding the experimental design. We fully agree that since both the overexpressed receptor (ADORA3 vs. P2RY12) and the host cell line (293T vs. A549) differ between conditions, observed differences in splicing cannot be definitively attributed solely to receptor-specific signaling, as they may also reflect inherent differences between the two cell lines.

In response, we have revised the manuscript as follows:

Added text in the manuscript(e.g., lines 49-77) stating that differential splicing outcomes between the ADORA3-293T and P2RY12-A549 models may arise from a combination of receptor-driven and cell-line-intrinsic effects, which cannot be disentangled in the present experimental design.

Systematically toned down the language in relevant conclusions. Claims that directly attributed splicing changes to GPCR signaling have been reframed to describe them as changes observed in the specific receptor/cell-line context, using more cautious and associative phrasing.

We appreciate the reviewer’s guidance on this important point, which has strengthened the accuracy and nuance of our data interpretation.

Comment 5: Absence of Short-Read Validation of DTU / AS Events

Although short-reads are used for FLAIR correction, they are not used to validate PSI values or confirm AS events.

Required revisions:

Validate at least 3–5 key AS events using short-read junction counts (e.g., sashimi plots).

Alternatively, explicitly state this limitation and mark the results as exploratory.

Respond 5:

Thank you for this critical comment regarding the short-read validation of AS events. In response, three sashimi plots, derived from short-read junction counts to illustrate AS events, have been incorporated into the supplementary materials.

Supplementary 2-1. Sashimi plot of ENST0000372813 and its isoform flairiso33189-1.

Supplementary 2-2. Sashimi plot of ENST0000380325 and its isoform flairiso50794-1.

Supplementary 2-3. Sashimi plot of ENST0000379161 and its isoform flairiso78895-1.

Comment 6: Figures Require Better Labeling and Clearer Interpretation

Examples:

Figure 2B: IGV screenshot lacks scale, track labels, and read depth.

Figures 3E–H & 4C–F: Dot plots are densely packed and difficult to interpret without color legends.

Figure 5: Novel isoform detection at KRT81 locus shown, but no quantitative read coverage or evidence.

Required revisions:

Improve figure resolution and readability.

Add clear legends, axis labels, and annotation tracks.

Provide more representative examples of splicing differences.

Respond 6:

We thank the reviewer for the valuable feedback on figure clarity and labeling. In response, we have implemented the suggested improvements: the resolution of all figures has been enhanced, clear axis labels and explanatory legends have been incorporated, and more representative examples of splicing differences have been provided. These revisions have been applied to ensure the data are presented with greater clarity and informativeness. We sincerely appreciate the reviewer's suggestions, which have been instrumental in improving the quality and readability of the figures.

Comment 7: Discussion Section Is Descriptive Rather Than Analytical

The discussion mostly restates results. It lacks:

Mechanistic hypotheses

Integration with relevant literature on GPCR-induced splicing regulation

Explanation of why A549 is more plastic beyond “it is a cancer cell line.”

Required revisions:

Add mechanistic interpretation tied to known splicing regulators.

Provide context: how do these findings compare to prior long-read datasets?

Expand on biological implications of alternative first/last exons (AF/AL), which are unexpectedly abundant.

Respond 7:

Thank you for these valuable suggestions. We have revised the whole discussion sections, added interpretation about the splicing mechanism, discussed the influence of second- and third-generatin sequencing, and explained why we observed so many AF/AL events.

Lines 396-449

AS is a major mechanism contributing to transcriptomic complexity and functional diversity [15]. However, its cell-type-specific characteristics, especially under genetic perturbations such as gene overexpression, remain insufficiently explored in commonly used cell models. In this study, we employed long-read RNA sequencing combined with FLAIR and SQANTI3 pipelines to profile the full-length transcriptomes of 293T and A549 cells under basal and GPCR-overexpressing conditions. Our findings reveal that while both ADORA3- and P2RY12-overexpressing cells exhibited significant global transcriptomic reprogramming, the nature and extent of these responses differed be-tween cell types. Our analysis revealed a substantial occurrence of alternative first/last exons (AF/AL), which contradicts previous findings suggesting skipped exons (SE) as the most prevalent alternative splicing (AS) event in human genomes [16]. Despite this, the quantity of genes affected by AF/AL is lower than that impacted by SE. Two hypotheses may account for this observation: first, the FLAIR-SQANTI3 pipeline might exhibit enhanced sensitivity to initial or terminal exon variations; second, it is plausible that AF/AL events are genuinely more frequent than SE events within cell line environments. Further research is necessary to empirically validate these postulations.

Alternative splicing has maintained its significance in biological research since the discovery of introns, primarily due to its pivotal role in fundamental processes like gene expression regulation [17-18]. Nevertheless, research in this area is hampered by the fact that the similarity between isoforms from the same gene can be confused with other factors, such as similarities among genes in the same family or other repetitive regions within genes [19]. Our prior research has demonstrated the viability of combining second- and third-generation sequencing methodologies for identifying novel plant isoforms, facilitated by the advent of third-generation sequencing in isoform characterization [20-22]. Similarly, third-generation sequencing has enabled the dis-covery of novel isoforms across diverse human tissues and cell types [23-25]. The identification of novel isoforms and gDTUs continues to face certain limitations. The utilization of only three replicates in this study, a consequence of resource restrictions, may exert an influence on the reliability of alternative splicing analysis, primarily because this analysis is acutely vulnerable to both biological noise and experimental batch effects. Furthermore, the inherent variability observed in ONT reads represents an often-overlooked aspect in the experimental design, a challenge that could be addressed through techniques like spike-in RNA quantification [26-27]. Thus, it is imperative to note that while these newly identified isoforms are supported by RNA-seq data, they necessitate additional experimental validation, such as PCR verification, prior to proceeding with functional investigations.

293T cells showed higher concordance with GENCODE annotations, a larger proportion of FSM, and fewer novel transcripts, suggesting a more stable and well-annotated transcriptomic baseline. In contrast, A549 cells displayed greater transcriptomic novelty and a broader transcriptional response to P2RY12 overexpression, consistent with their cancer-derived origin and inherent transcriptional plasticity [28]. DEG and gDTU analyses revealed distinct functional signatures. In 293T cells, DEGs and gDTUs were enriched in RNA processing and spliceosome-associated functions, implying that splicing regulation itself is a major target of transcriptional modulation in this context.

Conversely, A549 DEGs and gDTUs were more prominently involved in pathways related to cell cycle, translation, and stress response, reflecting the proliferative and stress-adaptive nature of cancer cells [29]. Additionally, ES emerged as the dominant AS event in both cell types, but the extent of gDTU was more pronounced in 293T cells. This suggests that 293T cells, despite their higher splicing fidelity, retain a capacity for fine-tuned splicing regulation in response to specific stimuli. The enrichment of spliceosome-related functions further positions 293T cells as a suitable model for mechanistic studies of splicing regulation. Given that this is a descriptive and exploratory study, mechanistic validation—such as RNA-binding protein (RBP) mapping or functional assays—will be required to determine the biological relevance of the identified transcript isoforms and splicing events [30].

Comment 8: English grammar, clarity, and conciseness

Several sections require language polishing for clarity (e.g., introduction and results are lengthy and repetitive).

Respond 8:

Thank you for pointing out these language issues. Accordingly, we have modified some parts and asked for assistance from the pre-English editing service for professional language editing. We hope this revised version meets the required standards.

Comment 9: Add a graphical abstract

Given the multi-dimensional analysis, a graphical summary would help readers.

Respond 9:

Thank you for your valuable suggestion regarding the addition of a graphical abstract. We fully agree that a graphical summary encapsulating the multi-dimensional analysis of the study would effectively help readers quickly grasp the core content and workflow of the research.

In accordance with your suggestion, we have redesigned the Figure 1. The new Figure 1 aims to integrate and summarize the core dimensions and key findings of the study in a plain way. We hope this addition could enhance the readability and presentation of the manuscript.

Comment 10: Methods need more explicit parameter reporting

Examples: FLAIR parameters, SUPPA2 ΔPSI threshold justification, SQANTI3 rescue criteria.

Respond 10:

Thank you for highlighting the need for clearer parameter reporting. We have added FLAIR parameters and SUPPA2 delta-PSI threshold justification in the revised methods section:

Lines 500-517

Long-read RNA sequencing data were processed using the FLAIR pipeline (version 2.0) following the recommended workflow. First, Illumina short reads from all samples were aligned to the hg38 reference genome using STAR with sjdbOverhang = 149, and splice junction files (SJ.out.tab) from each sample were merged to generate a comprehensive high-confidence splice-junction set for downstream correction.

Nanopore long reads were aligned to the hg38 genome using flair align with default minimap2 parameters. Splice-site correction was performed using flair correct, which incorporates both the STAR-derived splice junctions and the NCBI RefSeq gene annotation to refine long-read splice boundaries.

Transcriptome assembly was conducted using flair collapse, with several stringent options enabled, including --stringent, --check_splice, and --annotation_reliant, to ensure high-confidence isoform structures and reduce potential false-positive novel junctions. Read-to-transcript assignment files were generated with --generate_map.

Isoform quantification across samples was performed using FLAIR Quantify based on the unified transcriptome reference. Differential transcript expression was assessed using flair diffExp, whereas differential alternative splicing was tested using flair diffSplice with the --test option. All analyses were performed on the merged transcriptome and consistent sample-specific count matrices.

Lines 542-549

To enhance the analytical rigor, we applied stringent filtering criteria to the alternative splicing events identified by SUPPA2, as detailed below: First, to ensure reliability across biological replicates, each alternative splicing event was required to be detected in at least two out of the three biological replicates. This consistency filter was implemented using an in-house script. Second, to retain events with substantial biological differences, statistical thresholds were applied: only events meeting |ΔPSI| ≥ 0.1 and p-value < 0.05 were retained. This dual-filtering strategy was designed to ensure both the reliability and the statistical significance of the results for downstream analysis.

These details have now been explicitly included in the revised Methods section.

Comment 11: PCA plots (Figure 3A) require confidence ellipses or sample labels

Currently, it is unclear if replicates cluster tightly.

Respond 11:

We appreciate the reviewer’s feedback regarding the clarity of replicate clustering in Figure 3A. To address this, we have added sample labels and overlaid confidence ellipses for each condition in the revised figure. This adjustment improves the interpretability of within-group reproducibility, and the updated version is now included in the manuscript.

Comment 12: Ensure consistent terminology

“gDTU” vs “DTU genes” vs “AS events” interchanged inconsistently.

Respond 12:

We sincerely thank the reviewer for raising this important point regarding terminology consistency. We agree that the interchangeable use of “gDTU,” “DTU genes,” and “AS events” was unclear. We have revised the text throughout the manuscript to ensure precise and consistent terminology, primarily by:

Replacing the non-standard term “gDTU” with the clearer descriptions “genes exhibiting differential transcript usage (DTU)” or “DTU genes”.

Carefully distinguishing between DTU genes (the outcome of comparative analysis) and the alternative splicing (AS) events that may underlie these changes.

Lines 285–288

Genes with differential transcript usage (gDTU) analysis comparing transcriptomes before and after overexpression stimulation identified genes exhibiting significant differential transcript usage. These changes were often driven by specific alternative splicing (AS) events, such as exon skipping or intron retention.

Lines 439-441

Conversely, A549 DEGs and gDTUs were more prominently involved in pathways related to cell cycle, translation, and stress response, reflecting the proliferative and stress-adaptive nature of cancer cells.

Comment 13: Tone down statements implying broadly applicable biological conclusions

e.g., “These findings reflect tumor-specific processes” should be “These findings may reflect…”

Respond 13:

Thank you for your valuable suggestion. Your comment regarding the need for more precise language is very pertinent and greatly instructive for our future scientific writing. Following your advice, we have carefully reviewed the entire manuscript and modified similar statements that implied overly broad or definitive biological conclusions to adopt a more cautious and qualified tone. Here we provide a few representative examples of these revisions:

Lines 29-31

These findings suggest that 293T cells may be a suitable model for investigating splicing regulation, while A549 cells could serve as a relevant system for exploring tumor-related transcriptome dynamics.

Lines 54-58

While these observations point to a plausible link between extracellular signaling and splicing control, systematic studies on GPCR-driven alternative splicing remain limited. A key unresolved question is whether these splicing effects are cell-type-specific, especially within the common overexpression models routinely used in signaling research.

Lines 221-224

Our analysis reveals differences in the cellular characteristics of the two lines. These variations encompass multiple biological domains, including, but not limited to, transcriptional regulation, cytoskeletal organization, and extracellular matrix interactions.

Comment 14: Citations need minor corrections

Some references are missing publication details or have formatting inconsistencies.

Respond 14:

Thank you for highlighting the issues with reference formatting and completeness. We have carefully reviewed and corrected the entire reference list according to the journal's guidelines, ensuring that all publication details are included and formatting is consistent throughout. We appreciate your attention to this important detail.

Round 2

Reviewer 1 Report

Comments and Suggestions for Authors

The comments and recommendations made have been addressed and the manuscript can be accepted in the present form for publication.